# Correct Identification of Cell of Origin May Explain Many Aspects of Cancer: The Role of Neuroendocrine Cells as Exemplified from the Stomach

**DOI:** 10.3390/ijms21165751

**Published:** 2020-08-11

**Authors:** Helge Waldum, Patricia G. Mjønes

**Affiliations:** 1Department of Clinical and Molecular Medicine, Faculty of Medicine and Health Sciences, Norwegian University of Science and Technology, 7006 Trondheim, Norway; 2Department of Pathology, St. Olav’s Hospital, Trondheim University Hospital, 7006 Trondheim, Norway; patricia.mjones@ntnu.no

**Keywords:** carcinogenesis, cell of origin, dormancy, ECL cell, gastrin, gastric cancer, hormones

## Abstract

Cancers are believed to originate from stem cells. Previously, the hypothesis was that tumors developed due to dedifferentiation of mature cells. We studied the regulation of gastric acid secretion and showed that gastrin through the gastrin receptor stimulates enterochromaffin-like (ECL) cell histamine release and proliferation. In animal and human studies, we and others showed that long-term hypergastrinemia results in ECL cell-derived tumor through a sequence of hyperplasia, dysplasia, neuroendocrine tumors (NETs), and possibly neuroendocrine carcinomas (NECs) and adenocarcinomas of diffuse type. Perhaps, other cancers may also have their origin in differentiated cells. Knowledge of the growth regulation of the cell of origin is important in cancer prophylaxis and treatment. Physiology plays a central role in carcinogenesis through hormones and other growth factors. Every cell division implies a small risk of mutation; thus mitogens are also mutagens. Moreover, metastasis of slow proliferating cells may also explain so-called tumor dormancy and late recurrence.

## 1. Highlights

In animals and humans, long-term hyper-stimulation with gastrin results in growth of its target cell, the enterochromaffin-like (ECL) cell, which develops via hyperplasia to tumors of different degrees of malignancy. The central role of gastrin and the ECL cell in gastric cancer most probably has implications for other cancers and gives rise to improved prophylaxis and treatment. By using the most sensitive methods available, tumors should be classified not only by organ but also by cell of origin.

## 2. Introduction

Among the hallmarks of cancer are the ability to invade other tissues, to metastasize to other organs not in direct continuity with the tumor and not obey normal regulation of proliferation. These properties are due to genetic changes in the cancer cells, most often due to mutations. Only cells able to divide can give rise to neoplasia. Benign tumors develop when the number set point of a cell type is increased. With time, new mutations occur, and the tumor cell is gradually transformed into a more malignant one. Thus, benign tumors imply an increased risk of developing into cancer. Most often, in cancer cells there are an accumulation of mutations, but sometimes a mutation affecting a key gene results in a cancer cell with few mutations. In this review, I will focus on the cell of origin of tumors, particularly that differentiated cells are able to divide and that not only stem cells may develop into tumors. Moreover, since each cell division by chance implies a small risk of mutation, every stimulation of proliferation is accompanied by an increased mutation risk. Knowledge of the regulation of proliferation of the cell of origin will therefore indicate the mechanism of the tumorigenesis. Furthermore, signal molecules including hormones (estrogen, testosterone, gastrin) thus become important in tumorigenesis. Finally, understanding that differentiated cells with slow proliferation may also explain the phenomenon of so-called dormant tumor cells.

## 3. Causes of Mutations

Every division represents a small but definitive risk of mutation, although most of the errors that occur during DNA replication are repaired [1]. Mutations may be due to inborn errors, direct mutagens (DNA damage due to irradiation or chemicals) and mitogens [2] which, by increasing the number of divisions as well as shortening the time of division, increase the mutation risk. Mutations playing a role in tumorigenesis due to division without hyperstimulation represent just a chance. Finally, virus infections (Epstein–Barr virus, hepatitis B and C virus, human papilloma virus) may induce neoplasia by the incorporation of viral nucleic acid into the host cell genome [3].

## 4. Genes of Importance in Neoplasia

Traditionally, genes involved in tumorigenesis have been divided into two groups, oncogenes, where the mutations increase the function of genes involved in the growth of the cell, and tumor suppressor genes, which normally control and regulate proliferation or another important cellular function. When both alleles are affected by a mutation resulting in a loss of function, the actual control is lost. Moreover, loss of function mutations in genes involved in DNA repair also play an important role in tumorigenesis [4].

## 5. Cell of Origin

Correct identification of the cell of origin is of the utmost importance in oncology. The prevailing theory has been that tumors develop from stem cells based upon similarities between stem cells and cancer cells [5]. In benign tumors, the maturation stops late, leading to a tumor phenotype close to the normal tissue, whereas in malignant tumors, the maturation stops early, leading to tumors where the cellular origin may be difficult to recognize (Figure 1).

Another view is that tumors may develop from any cell able to divide [6]. With the accumulation of mutations, the differentiated cell gradually loses specific traits (secretory granules, synaptic markers and receptors leading to cells increasingly different from the cell of origin (Figure 1 and Figure 2).

Why is it so important to correctly identify the cell of origin? If the tumors develop from a differentiated cell, the growth regulation of this cell type becomes essential in tumorigenesis, and vice versa—if the tumors originate from the stem cell, its regulation becomes crucial. Thus, the receptors related to proliferation on the cell of origin will give an indication of the tumorigenesis. We will use our experience with the role of gastrin and the ECL cell in gastric tumorigenesis [8] to exemplify this question.

## 6. Gastric Cancer

Gastrin is recognized as a central regulator of gastric acidity [9]. Besides stimulating acid secretion, gastrin has long been known to have a positive trophic effect on the oxyntic mucosa [10]. It is now well-established that gastrin augments gastric acid secretion indirectly by stimulating release of histamine from the ECL cell [11]. Moreover, the gastrin receptor is localized to the ECL cell and probably not the parietal cell [12]. Gastrin has a specific positive trophic effect on the ECL cell as well as having a general trophic effect on the oxyntic mucosa [13]. Whether the general trophic effect is a direct one, mediated by a hitherto not recognized gastrin receptor on the stem cell, or indirect one by signal substances released from the ECL cell, including histamine [14] and Reg protein [15], is not solved. However, in rats we found that dosing with omeprazole (proton pump inhibitor (PPI)) induced trophic effects on the oxyntic mucosa in contrast to loxtidine (insurmountable histamine-2 antagonist), despite almost similar hypergastrinemia and identical increase in ECL cell density [16]. These findings suggest that the general trophic effect of gastrin on the oxyntic mucosa is mediated by histamine interacting with a histamine 2 receptor on the stem cell. However, in another study, loxtidine induced ECL cell hyperplasia and also, but to a lesser degree, a trophic effect on the oxyntic mucosa [17], indicating that the general trophic effect of gastrin on the oxyntic mucosa was not solely due to histamine acting via a histamine-2 receptor. Whether this effect on the oxyntic mucosa reflects an indirect effect, mediated for instance by Reg protein (14), or is due to a gastrin receptor on the stem cell remains to be determined. Very recently, Sheng et al. described gastrin receptor (CCK2R)-positive cells in the isthmus area of the oxyntic glands known as the localization of the stem cell [18]. Anyhow, gastrin is central in tumorigenesis in the oxyntic mucosa (Figure 3) [8,19]. ECL cell neoplastic transformation

The role of gastrin in the pathogenesis of ECL cell-derived neuroendocrine tumors (NETs) was recognized around 1980 [20,21]. At that time, the role of neuroendocrine cells in gastric carcinomas had been under consideration, primarily the ECL cell. [22]. It was realized that the distinction between carcinomas and carcinoids sometimes could be difficult [23]. After the introduction of potent inhibitors of gastric secretion, it became apparent that the tumorigenesis started with hyperplasia of ECL cells [24,25]. The hyperplastic ECL cells gradually transformed into NETs. Furthermore, NETs may develop further into highly malignant neuroendocrine carcinomas (NECs) [7,26] (Figure 1). In patients with hypergastrinemia due to atrophic gastritis localized to the oxyntic mucosa (autoimmune gastritis), we could demonstrate neuroendocrine differentiation of the tumor cells in gastric carcinomas when applying immunohistochemistry with increased sensitivity [27]. The normal ECL cell proliferates very slowly, causing some to deny their ability to divide [28]. The proliferative capacity of the ECL cell is now accepted [29]. Furthermore, we have repeatedly shown neuroendocrine/ECL cell differentiation in gastric carcinomas, especially of the diffuse type and particularly of the signet ring cell subtype [26,27,30,31,32,33,34,35,36] The transition of ECL cell hyperplasia to ECL cell NETs, particularly to NECs and adenocarcinomas in humans, has been disputed [37]. The classification of gastric carcinomas of diffuse type as adenocarcinomas, despite no glandular growth pattern, is partly based upon periodic acid-Schiff (PAS) positivity regarded as a marker of mucin. However, when applying much more sensitive and specific methods like immunohistochemistry and in situ hybridization, we could not find any mucin in the PAS positive cancer cells [35,38]. Thus, the ECL cell, a well differentiated cell, can divide, and it is possible to follow the tumorigenic process via hyperplasia to rather benign NETs, highly malignant NECs, and further to some carcinomas hitherto classified as adenocarcinomas. Moreover, the central role of gastrin is demonstrated by the fact that the most important cause of gastric cancer, *Helicobacter pylori* infection, predisposes to gastric cancer only after having induced oxyntic atrophy [39], which necessarily leads to hypergastrinemia. In fact, the pathogenetic mechanism for the carcinogenic effect of *Helicobacter pylori* gastritis may be hypergastrinemia [40]. Two years ago, the first report of increased occurrence of gastric carcinomas in patients using proton pump inhibitors (PPIs) compared with those stopping such treatment after *Helicobacter pylori* eradication, was published [41]. As gastrin is most probably the tumorigenic factor both for *Helicobacter pylori* gastritis and treatment with inhibitors of gastric acid secretion, an additive effect [42] explains the relatively short time period for the carcinogenic effect by PPI treatment in these patients [41]. Accordingly, identification of the cell of origin leads to an increased understanding of gastric carcinogenesis. Moreover, since we already have an efficient and safe gastrin antagonist, netazepide [43,44], this compound may be used to prevent cancer development. The gastrin receptor may be detected in an appreciable part of gastric carcinomas [36], indicating that netazepide could have a therapeutic effect in some gastric carcinomas. Very recently, it was reported that the decline in gastric cancer that had occurred for decades showed a break in early onset gastric cancer in the USA in about 1995 and thereafter started to increase [45]. Interestingly, the proton pump inhibitors (PPIs) had been introduced a few years earlier [46]. Moreover, in contrast to gastric mucosal lymphoma which is cured by *Helicobacter pylori* eradication and is not associated to oxyntic atrophy [47], patients with oxyntic atrophy due to *Helicobacter pylori* infection may develop gastric carcinomas many years after eradication [48], indicating that the carcinogenic process continues at this stage without the presence of *Helicobacter pylori* or inflammation. Patients with oxyntic atrophy have, however, reduced gastric acidity, causing hypergastrinemia, which may also drive the cancer development at this stage [49]. These patients probably would profit from a gastrin antagonist like netazepide [44]. *Helicobacter pylori* infection most often starts in childhood [50], whereas gastric cancer is generally a disease of middle-aged or old people, demonstrating a long latency. The debut of gastric cancer decades after *Helicobacter pylori* eradication, but with oxyntic mucosal atrophy without inflammation, is best explained by a role of hypergastrinemia in the carcinogenic process. Accordingly, eradicating *Helicobacter pylori* before development of oxyntic atrophy will most probably restore a normal stomach and remove increased risk of gastric cancer. If there is not complete oxyntic atrophy *Helicobacter pylori* eradication may reduce the risk, whereas at complete oxyntic atrophy *Helicobacter pylori* cannot live, and a gastrin antagonist when available could be tried in young patients with hypergastrinemia. Similarly, young patients with autoimmune gastritis may be treated with gastrin antagonist.

Gastrin immunoassays were developed around 1970. At that time gastritis was very prevalent, and many persons with gastritis were included in establishing the normal range of gastrin. However, since gastritis elevates gastrin, the normal upper level became too high [42]. Furthermore, there is no threshold for the tumorigenic effect of gastrin [51]. Both *Helicobacter pylori* infection and PPIs treatment alone or in combination may induce a degree of hypergastrinemia that in the long-term is sufficient to evoke gastric tumors based upon the gastrin values in patients with autoimmune gastritis and gastric NETs (carcinoids) [52]. Finally, to be complete, it should be stressed that gastrin at normal concentrations is an essential hormone regulating gastric acidity and also oxyntic integrity [53]. Moreover, at early phases of *Helicobacter pylori* infection only affecting the antrum, there is only a very slight increase in gastrin caused by NH_3_ due to urease [54] and possibly via methylhistamine [55].

## 7. Hormonal Carcinogenesis in General

Hormones have hitherto been regarded as incomplete carcinogens, meaning that they do not initiate the tumors, but that they can stimulate their growth when initiated. From what is written above, it should be evident that gastrin is a complete carcinogen for the ECL cell, and thus for gastric cancer. Similarly, estrogen was for many decades reported to predispose to vaginal cancers in girls born to mothers treated with estrogen in the hope of preventing spontaneous abortion [56]. Likewise, the large differences in the prevalence of breast cancer between the two sexes also reflects the dominating role of estrogen in this type of cancer. Furthermore, androgens play a central role in cancer of the prostate, which is utilized in the treatment of this cancer. Thus, long-term hyperstimulation of proliferation by hormones predisposes to cancer of the target cell, and hormones are therefore complete carcinogens. The denial of hormones as complete carcinogens seems to be due to neglection of the fact that mutations occur by chance, more often in conditions with accelerated cell division.

## 8. The Role of Neuroendocrine (NE) Cells in Carcinogenesis

Previously in this overview we have described the central role of the ECL cell in gastric carcinogenesis. The very slow proliferation of the ECL cell in the basal state implies that even a moderate augmentation nevertheless results in a rather slow growth taking years or decades to develop into an overt tumor [57]. A property of the ECL cells which may play a role in tumorigenesis is their production of histamine, which through its vascular effects [14] could promote cancer cell access to the blood and thus metastasis. We have previously also reported that the ECL cells do not express E-cadherin [58], which could explain their ability to spread in the tissue parallel to hereditary gastric cancer [59]. In general, the fact that the normal neuroendocrine cells occur as spread among other epithelial cells reflects that the adherence is weak between these cells and could indicate that they are prone to invade and spread. It is an old fact that neuroendocrine tumor cells may metastasize despite cytologically looking rather benign. Thus, these cells, due to their normal properties, do not need so many genetic changes before developing into a tumor. Unfortunately, our study on the expression of E-cadherin in normal neuroendocrine cells [58] has not been repeated by others. However, in pulmonary neuroendocrine tumors, E-cadherin was found in most tumors, but often with changed distribution [60], and an alteration of E-cadherin expression in neuroendocrine tumors of the gastrointestinal tract has also been reported [61].

Similarly, the EC cell produces serotonin, which also could facilitate tumorigenesis through its vascular effects. Accordingly, hormones and NE cells are generally much more important in tumorigenesis than presently recognized [62]. In that review, we described how the local neuroendocrine cells could be important in the carcinogenesis in organs like the lungs and pancreas. This point of view also explains why some neuroendocrine cells are more prone to developing into tumors than others, due to their normal production and secretion of substances affecting nearby tissue including blood supply and vascular bed. To differentiate between adenocarcinomas and NECs only by the percentage of tumour cells positive for a neuroendocrine marker is not logical (Box 1) since it depends on the sensitivity of methods applied.

Box 1Outstanding Questions.In malignant tumors, some more differentiated cells, often of neuroendocrine type, may be detected. When only scarce, these cells are neglected in the classification of the tumor, whereas when occurring above a certain limit, they may give rise to a classification as a neuroendocrine (NE) tumor. From a biological point of view, this seems peculiar. However, when applying methods with increased sensitivity, the number of cells showing the NE marker may increase, indicating that the tumor was a NE tumor. The distinction between adenocarcinomas and NE carcinomas should be explored more thoroughly and the NE cells should be subclassified based upon type of hormone produced.Based upon the knowledge of the cell of origin and its functional and trophic regulation (generally a close correlation between these effects), compounds with a negative effect should be tried in the treatment.The specificity of markers should be explored thoroughly.

## 9. Tumor Dormancy

A correct classification of the cell of origin may also help the understanding of the concept of tumor dormancy. Tumor dormancy, or quiescent cancer cells [63], is characterized by a short or long phase without any symptoms or signs of cancer in a person apparently cured of cancer, but who nevertheless harbors cancer which finally manifests itself by recurrence. Slow-cycling cancer cells, which often are more resistant towards cytotoxic drugs and thus survive such treatment, have recently been suggested to be the source of these dormant tumor cells [64]. Typically, neuroendocrine neoplasms of the less malignant type (NETs) [65], renal carcinoma [66] and breast cancer [67] may recur after years with apparent cure. There seems to be a correlation between the ability to early metastasize and the possibility of long-term dormancy [68]. The cells of origin of NETs are neuroendocrine cells, having such a slow proliferation that their ability to divide was denied [28]. It is also well-known that NETs without overt cytological changes and at small size may metastasize [69]. Regarding NETs, it was recently reported that nonsteroidal anti-inflammatory drugs had an antiproliferative effect on such tumors and therefore are possibly useful in treatment [70]. It should also be added that ghrelin-positive cells are found in gastric NETs of ECL cell origin [71], and that the enzyme ghrelin-O-acetyl transferase (GOAT) possibly may be used as a biomarker in the future [72]. Furthermore, we have previously focused on the clinical similarities between clear cell renal cancer and NETs and have shown that virtually all clear cell renal cancers express erythropoietin [73]. Based upon the latter finding, we proposed that the erythropoietin-producing cell could be the cell of origin of clear cell renal cancer with hypoxic-inducing factor (HIF) as an important growth regulator [74]. The cell of origin of breast cancer is still debated, but also in these cancers there are some with neuroendocrine differentiation [75], and recently markers common with neural crest were described in one subtype of cancer [76]. When a mutation occurs in a differentiated cell type prone to metastasis (such as a NET cell), new mutations occur in daughter cells, creating a heterogenous collection of cells with different degree of malignancy, all having the ability to metastasize which they also do. Thus, there is metastasis of differently malignant cells. The most malignant of them may be killed by cytotoxic drugs, whereas others are resistant and often called cancer stem cells [77]. However, these cells may be among the most benign ones with slow proliferation and thus recur after many years, even decades. Therefore, these tumor cells are not dormant but appear so because of their slow growth. Accordingly, our point of view is that the “quiescent” cells have not stopped proliferating, but that they have always been slow proliferators. With time, and after many cell divisions, these cells become more malignant with more rapid proliferation and they manifest themselves as new tumors identified as metastasis.

## 10. Conclusions

The crucial point in oncology is to identify the cell of origin. Through the knowledge of the normal growth control of this cell, it is possible to improve the prophylaxis, prevention and treatment of cancers. The heterogeneity of mutations in tumors, originating from slow proliferating differentiated cells, gives an explanation of so-called cancer stem cells and late recurrence. Hormones are central in growth regulation and thus carcinogenesis, being complete carcinogens. Carcinogenesis in many situations reflects physiology, and the normal functions of the cell of origin influence the behavior of the cancer. Tumor classification should generally be improved by being more biologically oriented. The most sensitive methods for markers having the highest specificity should be applied in tumor classification. Thus, tumors should not only be classified according to organ, but also by cell of origin.

## Figures and Tables

**Figure 1 ijms-21-05751-f001:**
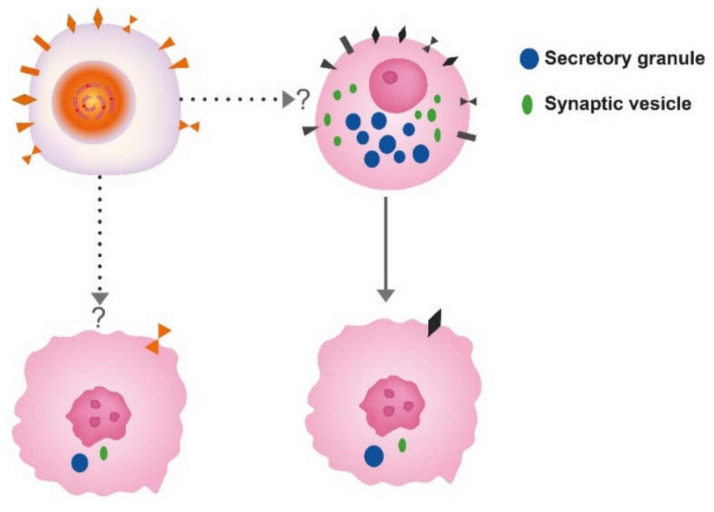
The different theories for the cell of origin of cancers; the stem cell upper left and differentiated cell, here depicted as a neuroendocrine cell to the right. Whether neuroendocrine cells in the gastrointestinal tract develop from the stem cell or separate neuroendocrine precursors is still debated. Anyhow, both cells may convert into cancer cells of similar phenotype.

**Figure 2 ijms-21-05751-f002:**
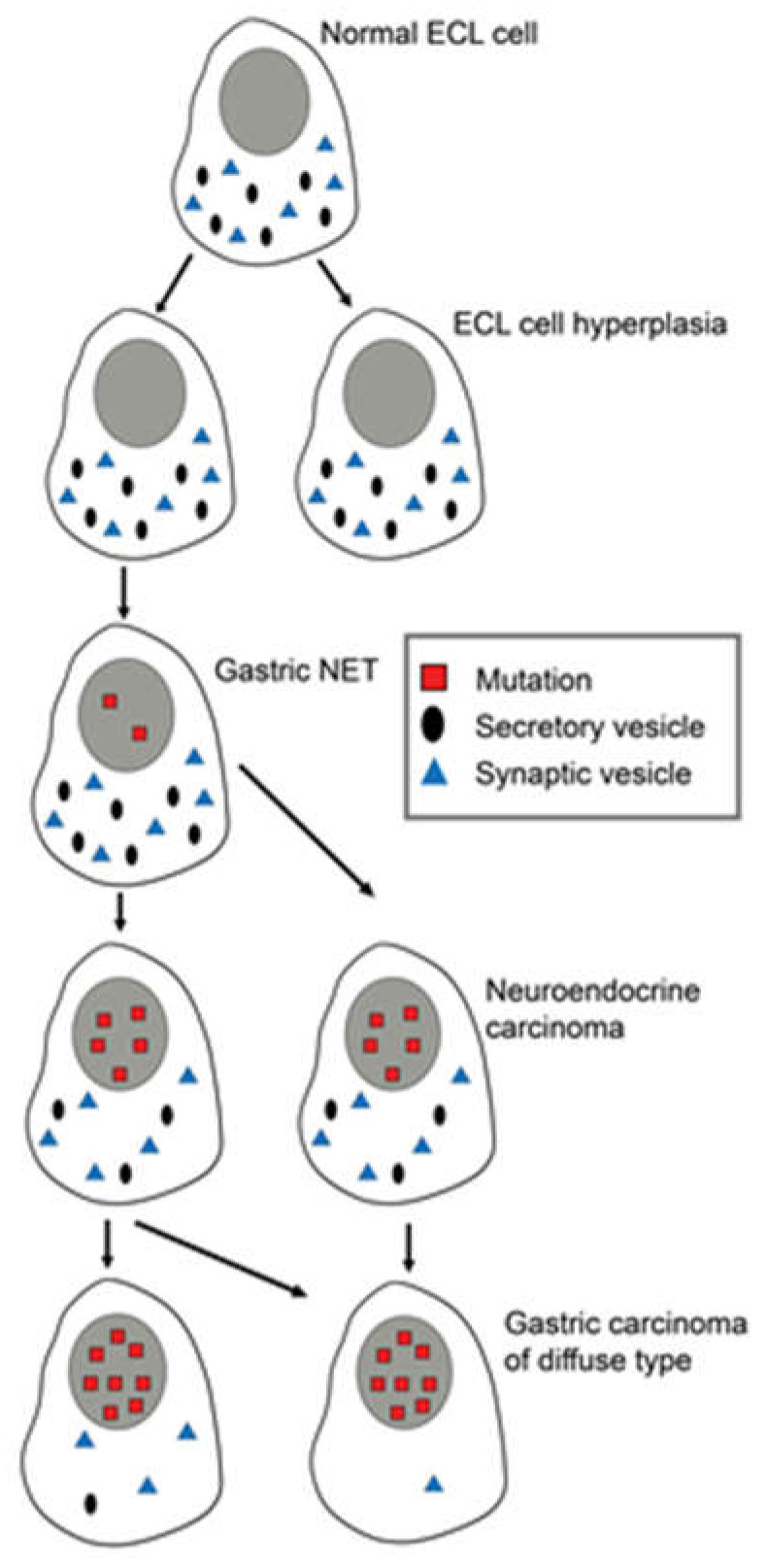
Changes in differentiated ECL cells during tumorigenesis/carcinogenesis with accumulation of mutations, leading to loss of specific markers, making it increasingly difficult to recognize the cell of origin. Reproduced with permission from Waldum HL, Brenna E, Sandvik AK 1998. Relationship of ECL cells and gastric neoplasia. Yale J Biol Med 71; 325–335 [7].

**Figure 3 ijms-21-05751-f003:**
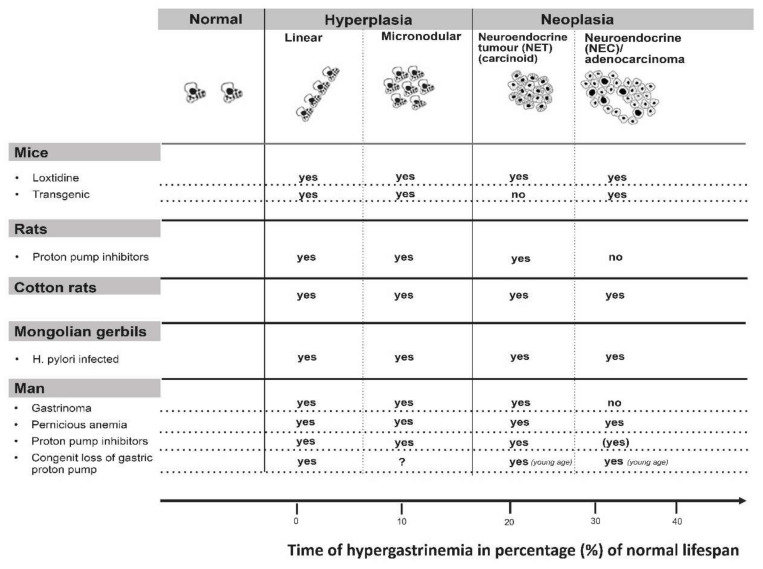
Every condition with long-term hypergastrinemia leads to ECL cell tumors in all species examined. Reproduced with permission from: Waldum et al. Expert Opinion. *Drug Safety*
**2002**, *1*, 29–38 [19].

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
