# Peer review of "Correct Identification of Cell of Origin May Explain Many Aspects of Cancer: The Role of Neuroendocrine Cells as Exemplified from the Stomach"

_ijms, 2020, doi:10.3390/ijms21165751_

Round 1
Reviewer 1 Report
General comment
This is an interesting review in which Dr. Waldum and coworker(s) based on more than four decades of and impressive experimental and clinical studies of gastric function and tumor development in the stomach summarize their results in a general, oncological conclusion. The argumentation is fairly convincing, and the information provided about the role of neuroendocrine cell biology should be of value in many aspects of basic cancer research.
Minor comments
- Maybe more direct information about the degrees and levels of hypergastrinemia during PPI-treatment, helicobacter infection and atrophic gastritis would be helpful for gastroenterologists.
- Line 168: The word “is” is missing in the last sentence.
- Line 289-290: The title in ref. 21 should probably not be written in capital letters.
Author Response
1.The level of gastrin is included, lines 182-188. To give exact values are difficult due to variability in the assays used.
2. "is" is included
3. The title of ref. is changed from capital to lower case.
Reviewer 2 Report
In this review, Authors have focused the attention on cancers originating from stem cells unlike the original hypothesis that tumors have developed due to dedifferentiation of mature cells. Authors who shared a lot of experience in gastrin research did concentrate on pathophysiology of gastrin, its receptors and effects on cell proliferation and ECL cells releasing histamine. From their own and other studies it is obvious that long-term hypergastrinemia results in ECL cell derived tumour through a sequence of pathomorphological events including hyperplasia, dysplasia, neuroendocrine tumour (NETs), neuroendocrine carcinomas (NECs) and adenocarcinomas of diffuse type. Interesting aspects of helicobacter pylori induced pathology of animal and human stomach as well as the problems met during experimental and clinical settings e.g. with metastasis, tumour dormancy and cancer recurrence are also presented.
In overall this is valuable review on very interesting topic, highly relevant to human gastric cancer and I only have couple suggestions for Authors to present more detailed recent evidences and improvement of this review in its original version.
Critical comments
- Page 2 lines 45; please add which hormones according to authors are important for the regulation of tumorigenesis?
- Page 2 line 60-61, please reveal which particular virus(es) can induce neoplasia?
- Figure 1 should be enlarged and the caption of better edited because in a format presented, is not easy readable.
- Page 3, lines 89-94, tumor promoting efficacy of gastrin is highlighted in this review. However the beneficial effect of this hormone when cited, are missing role of gastrin in gastric integrity and gastroprotection. Authors should discuss earlier observation regarding this protective activity of this peptide published in Eur J Pharmacol. 1995; 24;278:203-12. and Gastroenterology 1995 ;109(1):89-97 which may account for the beneficial effects of this peptide.
- The histamine metabolite methylhistamine should be mentioned when discussing the carcinogenic effect of Helicobacter pylori (page 5) in order to emphasize the pathogenicity of this bug.
- Page 6, the relation of N-cadherin and E-cadherin in reference to ECL cells is very interesting in terms of their spreading along with gastric cancer, although this is unclear from the text of this paper. Please refer to this issue in more detailed fashion.
- Authors may point out to the recent developments in NETs pathogenesis and potential biomarkers when discussing tumor dormancy (page 6). For instance, studies on the therapeutic role of NSAIDs and COX-2 (PMID: 30420588) and ghrelin-O-acyltransferase (GOAT) (PMID: 30297816) seem interesting and should add more tasty flavor to this paragraph.
Author Response
Comment 1: Page 2, line 45-46. Comment 2: Page 2, line 54-55. Comment 3. Figure 3 was already changed. Comment 4. Page 6, line193-197. Comment 5. Page 6, line196-197. Comment 6. Page 7, line224-228. Comment 7. Page 7, line249-255. Furthermore, a comment on a very recent paper by Sheng et al. is included on page 4, line 113-114. Nine references are included.Reviewer 3 Report
The authors review and provide their point of view, based on their previous experience in this field and reviewing the publications, regarding the cell of origin of cancers. Globally, this is a review and one more of work in this field that discusses one matter that is still very much under debate.
some minor comments:
- Define ECL cell for the general readers that might not know this abbreviation
- The transition of NEC hyperplasia to NET and then NEC is not the agreement of all researchers. It is better that authors point out some other publications that do not support such a transition.
- Point 2 becomes more important when thinking about NEN globally and not limited to GI tract but in entire body. How the authors define the cell of origin of lung SCC or ADC? The same as NEN of lung? Could the hypothesis of authors being generalized to all organs? And if not, what is the position of authors to this question?
- Mixed features of ADC and NEC and even some morphological changes in metastases of the primary tumor are reported but these features are not strong enough to fully support the idea of ECL being the cell of origin of all types of malignancies in different organs. Do the authors focus on GI cancers or try to generalize their point to all other organs? are seen but this feature is not enough
- What authors mean by their statement if Figure 1, that the NEC are the only MUCOSAL cells besides the stem cells, having the ability to proliferate? All gastric mucosal cells proliferate.
Author Response
- ECL cell is defined: line 27
- The different views on the role of neuroendocrine cells in tumourigenesis is mentioned and the long-term opponent against this view, E. Solcia is included as a new reference. Lines 137-139.
- We are sorry that we obviously have been unclear concerning the role of neuroendocrine cells in tumourigenesis in other organs. We have referred to our previous article where the role of other neuroendocrine cells occurring normally in other organs may play a similar role in tumourigenesis. We focused in the previous review particularly on the pancreas and the lungs. Lines: 218-223.
- Our view is that the neuroendocrine cell in an actual organ can play a role in tumourigenesis. The ECL cells plays a role only in the stomach.
- Thank you for your remark. The figure legend is changed. Lines 72-75
Reviewer 4 Report
The review work by Waldum et Mjones highlights the possible derivation of several malignant tumors from differentiated cells, often of neuroendocrine origin. When only scarce, these cells are neglected in the classification of the tumors even if the precise distinction between adenocarcinomas and neuroendocrine components should be explored more deeply and extensively in order to possibly improve prophylaxis, prevention and treatment of cancers. The authors extensively describe the example of stomach ECL cells in gastric neoplasia progression.
The review is clear written and reports appropriate references. According to the title I would expect to find more information about general mechanisms; instead the manuscript is mainly focused on gastric cancer. Probably a more specific title would be appropriate to represent and recapitulate better the review contents
If possible, the authors should enlarge of Figure 1 and 3 because the fonts are illegible. They could also provide original illustrations or tables to better summarize the chapters and highlight the conclusions.
Author Response
1.The title has been extended with a subheading. Line 3.
2.Figure 3 is replaced with a new version and figure 1 is improved.